# It Takes Two to Tango: A Bacterial Biofilm Provides Protection against a Fungus-Feeding Bacterial Predator

**DOI:** 10.3390/microorganisms9081566

**Published:** 2021-07-23

**Authors:** Shubhangi Sharma, Stéphane Compant, Philipp Franken, Silke Ruppel, Max-Bernhard Ballhausen

**Affiliations:** 1Leibniz Institute of Vegetable and Ornamental Crops, Theodor-Echtermeyer-Weg 1, 14979 Großbeeren, Germany; sharma@erfurt.igzev.de (S.S.); philipp.franken@fh-erfurt.de (P.F.); Ruppel@igzev.de (S.R.); 2AIT Austrian Institute of Technology, Center for Health and Bioresources, Konrad Lorenz Strasse 24, 3430 Tulln, Austria; stephane.compant@ait.ac.at; 3Institute of Microbiology, Friedrich Schiller University Jena, Neugasse 24, 07743 Jena, Germany

**Keywords:** bacterial-fungal interactions, biofilm, mycophagy

## Abstract

Fungus-bacterium interactions are widespread, encompass multiple interaction types from mutualism to parasitism, and have been frequent targets for microbial inoculant development. In this study, using in vitro systems combined with confocal laser scanning microscopy and real-time quantitative PCR, we test whether the nitrogen-fixing bacterium *Kosakonia radicincitans* can provide protection to the plant-beneficial fungus *Serendipita indica*, which inhabits the rhizosphere and colonizes plants as an endophyte, from the fungus-feeding bacterium *Collimonas fungivorans*. We show that *K. radicincitans* can protect fungal hyphae from bacterial feeding on solid agar medium, with probable mechanisms being quick hyphal colonization and biofilm formation. We furthermore find evidence for different feeding modes of *K. radicincitans* and *C. fungivorans*, namely “metabolite” and “hyphal feeding”, respectively. Overall, we demonstrate, to our knowledge, the first evidence for a bacterial, biofilm-based protection of fungal hyphae against attack by a fungus-feeding, bacterial predator on solid agar medium. Besides highlighting the importance of tripartite microbial interactions, we discuss implications of our results for the development and application of microbial consortium-based bioprotectants and biostimulants.

## 1. Introduction

The rhizosphere constitutes the hot-spot for soil microbial activity where bacteria and fungi coexist and interact [1,2]. Plants drive microbial communities in the rhizosphere by establishing a continuous flow of carbon mainly in the form of rhizodeposits [3,4,5,6] which comprise sugars, amino acids, and organic acids [7,8]. More than 10% of the C transported below ground is allocated to mycorrhizal or saprophytic fungi and to rhizobacteria, which constitute the bulk component of the rhizosphere but can also colonize plants as endophytes [3,9,10,11,12,13,14].

These fungi act as secondary carbon sinks in the soil and can recruit distinct bacterial communities via modulating the composition of their exudates leading to the so-called “mycorrhizosphere” and “sapro-rhizosphere” [15,16,17,18]. These fungi-associated bacteria can feed on fungus-derived exudates, in exchange for beneficial services provided to the fungi [16,19,20]. In addition to this, some of the bacteria inhabiting the myco/saprorhizosphere can form tight physical associations with the fungi which can often lead to the formation of biofilm-like structures [21,22,23,24]. Biofilms are described as the aggregation of bacterial cells embedded into a self-produced matrix of extracellular polymeric substances [25]. These biofilms can have beneficial effects, including nutrient exchange between both partners and the migration of bacteria along the hyphae [24,26,27]. Regardless of the effect of bacterial biofilms on the fungal partner, the existence of a biofilm lifestyle is rather common in nature [28]. Furthermore, biofilm formation on the fungal hyphae can provide the fungal partner protection against predators, toxic compounds, and buffering of environmental variations [29,30]. However, bacterial biofilms on fungal hyphae can also compete for nutrients and this can be detrimental for the fungi [31,32].

Some soil-inhabiting bacteria have evolved different strategies to compete for limited resources in the myco/saprorhizosphere. They can either obtain C from the exudates released by fungi or by using the fungal hyphae as a substrate [33,34,35]. The existence of fungivorous bacteria which can feed on living fungal tissue has been extensively studied for bacteria belonging to genus *Collimonas* [36,37]. Collimonads can feed on living fungal biomass and convert it into bacterial biomass, using a strategy known as “mycophagy”. These bacteria are known to have antifungal properties [36,38,39] and have been extensively studied for their antagonistic activity against pathogenic soil fungi [36,39,40,41,42,43,44,45,46,47], saprophytic fungi [48], and beneficial fungi [38,49], thereby emphasizing their role in altering the fungal community composition in the soil ecosystem [50,51].

As opposed to feeding on fungal hyphae or intrahyphal contents, a large number of soil bacteria have evolved strategies to utilize fungus-secreted metabolites [52,53]. Bacteria belonging to the genus *Kosakonia* of the family *Enterobacteriaceae* are common residents of the rhizosphere which are often found to be associated with plant tissues as endophytes and can promote plant growth via different mechanisms [54,55,56,57]. *Kosakonia radicincitans* (DSM 16656) is a nitrogen-fixing bacterium [58], which has a broad host range and upon successful inoculation can promote plant growth and yield under greenhouse conditions and field trials [59,60,61,62]. This makes it a potential bio-supplement suitable for application in agricultural systems. However limited information is available on the interaction of *K. radicincitans* with common soil fungi [63].

*Serendipita indica*, previously known as *Piriformospora indica*, is an AM-like plant-beneficial root-endophytic fungus belonging to the family *Serendipitaceae* (formerly Sebacinales group B) [64]. This is a ubiquitous fungus that has been isolated from different ecological environments [65,66] and is known to colonize the roots of diverse plant species [67,68]. It can improve plant growth via a wide range of mechanisms, such as increasing nutrient uptake, inducing tolerance against biotic and abiotic stresses, and conferring resistance against pathogens [67,69,70,71]. However, few studies have highlighted the outcome of its interaction with rhizospheric bacteria [70,72].

The interaction of collimonads with pathogenic fungi has been well documented to support their role as potential biocontrol bacteria [42,73,74]. However, the impact of these bacteria on common soil fungi still remains largely unknown. Interestingly, mycophagous bacteria have been found to be associated with saprotrophic fungi such as *Mucor hiemalis* and *Trichoderma harzianum* and plant-beneficial fungi such as the AM fungi *Rhizoglomus irregulare* (previously known as *Glomus intraradices*) and *Gigaspora decipiens,* and the ectomycorrhizal fungus *Laccaria bicolor* [75,76,77]. In addition to this, the mycophagous bacterium *Collimonas pratensis* was recently isolated as an endophyte inhabiting the mycorrhizal tissues of orchid plants [78]. Altogether, this indicates that mycophagous bacteria are widespread in nature across different ecological habitats. Höppener-Ogawa and colleagues [51] reported that collimonads can alter the composition of common soil fungi in the rhizosphere including plant-beneficial AM fungi. However, how this impacts the biomass of beneficial fungi in the rhizosphere is not clear as the authors measured the total soil ergosterol, which is not a suitable biochemical marker for estimating the biomass of AM fungi [79,80,81]. Interestingly, the presence of collimonads also altered the fungal community structure of AM fungi in the plant roots. This is a particularly important result as it points out that even though collimonads are wide-spread in terrestrial environments at low densities (10^4^–10^6^ cells per gram of soil) they can still strongly impact the fungal community composition in the soil ecosystem. In addition to this, studies have shown that collimonads have a feeding preference for certain specific taxonomic or functional groups of fungi [41,50,51]. Therefore, it is imperative to know whether collimonads can feed on beneficial fungi such as *S. indica* leading to a potential loss of fungal inoculum in the fields. It is also important to test whether bacterial-fungal consortia comprising of biofilm-forming bacteria can protect the fungus against attack by mycophagous bacteria.

Therefore, in the present study we tested whether *K. radicincitans* can confer protection to the beneficial root-endophytic fungi *S. indica* against fungus-feeding bacterium *C. fungivorans* by forming a biofilm on the fungal hyphae. We conducted in vitro mycophagy assays to evaluate whether the two bacteria can feed on fungal biomass, and investigated biofilm formation and different feeding strategies used by the bacteria to benefit from *S. indica*.

## 2. Materials and Methods

### 2.1. Microorganisms Used in This Study

In the present study we used the root-endophytic fungus *Serendipita indica* (DSM 11827) which was originally isolated from the Thar Desert in India [82]. In addition to this, the two bacteria used in this study include the mycophagous bacterium *Collimonas fungivorans* (Ter 331) (Genbank accession numbers AJ310395 and CP002745.1 for sequenced 16SrDNA and whole genome, respectively) which was isolated from slightly acidic dune soil in the Netherlands [36]. The nitrogen-fixing bacterium *Kosakonia radicincitans* (DSM 16656) was isolated from the phyllosphere of wheat plants [58].

### 2.2. Cultivation of Microorganisms

*S. indica* was routinely cultured on Hill-Käfer complex medium [83] (CM; glucose 20 gL^−1^, Peptone 2 gL^−1^, yeast extract 1 gL^−1^, casamino acid 1 gL^−1^, agar 15 gL^−1^, 50 mL 20× salt solution (NaNO_3_ 120 gL^−1^, KCl 10.4 gL^−1^, MgSO_4_·7H_2_O 10.4 gL^−1^, KH_2_PO_4_ 30.4 gL^−1^), 10 mL microelements (MnCl_2_·4H_2_O 0.5 gL^−1^, H_3_BO_4_ 1.4 gL^−1^, ZnSO_4_·7H_2_O 2.2 gL^−1^, FeSO_4_·7H_2_O 0.5 gL^−1^, (NaH_4_)_6_Mo_7_O_24_·4H_2_O 0.11 gL^−1^, CuSO_4_·5H_2_O 0.16 gL^−1^, CoCl_2_·5H_2_O 0.16 gL^−1^, Na_2_EDTA.2H_2_O 5.1 gL^−1^; pH 6.5) using standard 90 mm Petri dishes and incubated at 24 °C for two weeks.

*C. fungivorans* and *K. radicincitans* were cultivated in 10% tryptic soy broth or solid medium (3 gL^−1^ TSB; Sigma Aldrich, Munich, Germany). The Petri dishes were incubated at 24 °C in a stationary incubator. The bacterial inoculum was prepared by transferring a single colony into an Erlenmeyer flask containing 50 mL of 10% TSB broth (Carl Roth, Karlsruhe, Germany). The flasks were incubated at 24 °C under gentle shaking at 180 rpm for all the assays. The cells were harvested by centrifugation (10,000× *g*, 10 min), rinsed three times in sterile NaCl (0.05 M) solution, and re-suspended in saline solution to a final concentration of 10^6^ CFU per mL for all in vitro experiments unless mentioned otherwise. Glycerol stocks of all microorganisms were maintained in 70% glycerol at −80 °C.

### 2.3. Fungal Growth Inhibition Assay

Assays were conducted on water yeast agar (WYA) (KH_2_PO_4_ 1 gL^−1^, NaCl 5 gL^−1^, yeast extract 0.05 gL^−1^ and agar 20 gL^−1^, pH 6.8) using standard 90 mm Petri dishes. This media imitates the carbon limiting conditions of the soil [84]. *C. fungivorans* (Ter 331) and *K. radicincitans* (DSM 16656) were grown in 10% TSB and incubated at 24 °C for 48 h at 180 rpm. Bacterial inoculum was prepared as mentioned in Section 2.2. 5 µL of bacterial suspension was inoculated on a Petri dish in a 40 mm × 40 mm square for 4 days before the fungus was introduced. *S. indica* was cultivated on Hill-Käfer complex medium. On the 5th day, a 10 mm plug from a two-week-old actively growing *S. indica* colony was placed in the center of the plate at a distance of approximately 20 mm from the bacterial patch. The plates were incubated at 24 °C for 3 weeks and were scored for fungal growth. The experiment was done in triplicate and repeated three times.

### 2.4. Mycophagy Assay

A mycophagy assay was conducted to test whether *C. fungivorans* (Ter 331) and *K. radicincitans* (DSM 16656) can feed on *S. indica* (DSM 11827). The assay was performed as described by Ballhausen and colleagues [48]. Briefly, *C. fungivorans* and *K. radicincitans* were grown overnight in 10% TSB at 24 °C. The bacterial suspension was prepared as described in Section 2.2. A Petri dish containing phytagel medium (Phytagel 4 gL^−1^ and MgSO_4_ 0.72 gL^−1^) (Sigma Aldrich, Munich, Germany) was inoculated with 100 µL of bacterial suspension using a sterile spreader. Phytagel forms a semi-solid medium which contains almost no nutrients [85]. A 10 mm fungal plug was excised from a two-week-old *S. indica* colony and placed on an autoclaved aluminum disc in the center of the Petri dish, to avoid the diffusion of nutrients from the fungal plug into the phytagel (Appendix A). The plates were incubated at 24 °C for two weeks. Phytagel plates with or without fungi/bacteria were used as controls. Hyphal morphology of *S. indica* was observed in all treatments using a stereomicroscope (Leica SZX 10, Wetzlar, Germany) after 2 weeks of incubation including uninoculated fungus. The experiment was done using four replicates and repeated three times. After incubation, the fungal plugs were removed carefully, the bacteria were washed with 2 mL MES buffer (pH 5.5), and the optical density (OD_600_ nm) of the bacterial suspension was measured and compared to the bacteria or uninoculated fungus controls. Mycophagy was scored as ratio of OD_treatment_/OD_controls_, quantified by dividing the average OD_600_ of each treatment by the average OD_600_ of the control (either bacteria or fungus only). The experiment consisted of four replicates. In order to retrieve fungal biomass, phytagel medium was dissolved using citrate buffer (8.3 mM trisodium citrate, 1.7 mM citric acid, and 1% Triton X100; pH 6.0) as described by Cranenbrouch et al. [86]. The Petri dishes containing fungal hyphae were incubated with 5 mL of citrate buffer for 2 h at 50 °C, and the fungal biomass was collected and washed with water to remove traces of phytagel. After drying at 40 °C for 24 h, fungal dry weight was measured.

### 2.5. Extraction of Metabolites of S. indica

*S. indica* (DSM 11827) was grown in liquid CM for two weeks at 24 °C under gentle shaking at 180 rpm. The fungal metabolites were extracted by adding Resin Amberlite XAD4 (Sigma Aldrich, Munich, Germany) at 40 gL^−1^ of Hill-Käfer complete medium. In order to recover the fungal metabolites from the resins, they were washed with 200 mL of methanol three times. The recovered metabolites were concentrated (470 times) via rotatory evaporator. The media without the fungus was extracted in the same way and used as the control.

### 2.6. Growth of the Bacteria on Metabolites of S. indica

In order to evaluate the growth of *K. radicincitans* (DSM 16656) and *C. fungivorans* (Ter 331) on *S. indica* (DSM 11827) metabolites recovered in Section 2.5, a growth assay was conducted using 96-well plates (Greiner, Frickenhausen, Germany). For this purpose, M9 minimal media was used as this medium contains only salts and nitrogen, and can be supplemented with a carbon source and vitamins as needed [87]. The M9 medium (Na_2_HPO_4_ 6.76 gL^−1^, KH_2_PO_4_ 3 gL^−1^, NaCl 0.5 gL^−1^, NH_4_Cl 1 gL^−1^, pH 6.8) was amended with fungal metabolites (see Section 2.5) as the only source of carbon (2% metabolites *v*/*v*). In the control treatment, M9 medium was amended with metabolites recovered from media without *S. indica* (see Section 2.5) as the sole source of carbon (2% metabolites *v*/*v*). A bacterial suspension of *K. radicincitans* and *C. fungivorans* was prepared as described in Section 2.2 and the cell density was adjusted to 10^6^ CFU per mL. The 96-well plate was inoculated with 170 µL of M9 media amended with *S. indica* metabolites or metabolites extracted from the CM without *S. indica* (see Section 2.5). Each well was inoculated with 10 µL of bacterial suspension. The plates were incubated at 24 °C at 120 rpm and growth of *K. radicincitans* and *C. fungivorans* was measured as optical density (OD_600_ nm) every 24 h for 3 days using a plate reader (Tecan 1100, Männedorf, Switzerland).

### 2.7. Co-Cultivation of Bacteria and Fungus (with or without Physical Contact)

*S. indica* (DSM 11827) was co-cultured either with *C. fungivorans* (Ter 331) or *K. radicincitans* (DSM 16656) using a 6-well plate (Sarstedt, Nümbrecht, Germany). In order to obtain a *S. indica* colony without agar, a cellophane membrane was used to culture the fungus as described [88]. Briefly, cellophane membranes were boiled in EDTA (1 gL^−1^ of deionized water) for 30 min, rinsed with water, and autoclaved at 120˚C for 15 min. Membranes were put on top of solidified CM in a 90 mm Petri dish. Using a sterile surgical blade, fungal mycelia were scraped from a two-week-old *S. indica* plate and placed on the cellophane membrane. The Petri dishes were incubated at 24 °C for 5 days to get a colony of appropriate size (10 mm × 10 mm). Cellophane membranes containing fungal colonies were washed in NaCl (0.08 M) solution with gentle shaking to release the fungal colonies (Appendix A). A bacterial suspension of *C. fungivorans* and *K. radicincitans* was prepared as described in Section 2.2. The cell density was adjusted to 10^6^ CFU per mL. Each well in the 6-well plate was inoculated with 3 mL of phosphate buffered saline solution (PBS; NaCl 8 gL^−1^, KCL 0.2 gL^−1^, KH_2_PO_4_ 0.24 gL^−1^, and Na_2_HPO_4_ 1.44 gL^−1^; pH 7.4) and 50 µL of bacterial suspension of *C. fungivorans* or *K. radicincitans* along with one fungal colony patch. To physically separate the fungus and bacteria, well inserts with a pore size of 0.4 µm (Sarstedt, Nümbrecht, Germany) were used (Appendix A). The bacteria were incubated in PBS, with the following treatments: (1) bacteria alone; (2) bacteria in physical contact with *S. indica*; (3) bacteria without physical contact with *S. indica*. The growth of the bacteria was measured as the bacterial optical density (OD_600_ nm) every 24 h for 3 days. The experiment was performed with six independent replicates and repeated twice.

### 2.8. Tripartite Interaction: K. radicincitans, S. indica and C. fungivorans on Solid Media

*S. indica* (DSM 11827) was cultivated on CM plates for 1 week. *K. radicincitans* (DSM 16656) was cultured on 10% TSB plates and incubated at 24 °C for 48 h. One single colony of *K. radicincitans* was carefully picked using a sterile loop and inoculated parallel to the fungal colony at 20 mm distance. After 3 days of co-cultivation, when fungal hyphae encountered *K. radicincitans*, a fungal plug of 10 mm was taken from the point of contact and placed on nutrient-limiting water yeast agar (WYA) plates. The bacterial-fungal consortium was allowed to grow on WYA plates for 1 week; during this time, *K. radicincitans* formed a thick biofilm on the fungal hyphae, which was observed using a stereomicroscope (Leica SZX 10, Wetzlar, Germany). *C. fungivorans* was grown as described in Section 2.2. The cell density was adjusted to 10^6^ CFU per mL. 10 µL of bacterial suspension was streaked parallel to the fungal colony at 20 mm distance. The plates were incubated at 24 °C for 10 days after which a photographic record of the bioassay was registered using a Nikon D750digital SLR camera (Nikon Inc., Sendai, Japan) using identical camera settings and light conditions. The growth of the fungus was measured as the surface of the Petri dish covered by the fungal mycelia using ImageJ 1.48 software (https://imagej.nih.gov/ij/, accessed on 22 June 2021). The experiment consisted of nine replicates.

### 2.9. Study of Tripartite Interaction: K. radicincitans, S. indica, and C. fungivorans in Liquid Media

In order to evaluate the protective effect of *K. radicincitans* on the growth of *S. indica* in the presence of the fungus-feeding bacterium *C. fungivorans*, an in vitro assay was conducted using 6-well plates. The experiment consisted of the following treatments: (i) *S. indica* co-cultivated with both *K. radicincitans* and *C. fungivorans*; (ii) *S. indica* co-cultivated with *K. radicincitans*; (iii) *S. indica* co-cultivated with *C. fungivorans*; and (iv) uninoculated *S. indica*. Briefly, fungal colonies of approximately 10 mm × 10 mm were cultivated as explained in Section 2.7. Bacterial suspensions of *K. radicincitans* and *C. fungivorans* were prepared as described in Section 2.2. The final cell density was adjusted to 10^6^ CFU per mL. In the case of treatment (i) (see above), each well was inoculated with one fungal colony and 100 µL of *K. radicincitans*. Thereafter, the 6-well plates were incubated at 24 °C for 3 h with no shaking (to allow the formation of biofilm). After 3 h, the 6-well plates were inoculated with 100 µL of *C. fungivorans* and the plates were incubated at 24 °C, with gentle shaking at 80 rpm to aerate the system. In the case of treatments (ii) and (iii) (see above), each well was inoculated with one fungal colony and 100 µL of *K. radicincitans* or *C. fungivorans*. In the case of treatment (iv) (see above), each well was inoculated with a single colony of *S. indica* and 100 µL of sterile saline solution (0.05 M). The plates were incubated at 24 °C, with gentle shaking at 80 rpm. Each treatment consisted of six biological replicates and three technical replicates. Samples were taken after 2 h and 72 h for DNA extraction.

### 2.10. DNA Extraction and Quantitative PCR (qPCR)

Total DNA was extracted from fungal colonies using the Qiagen DNeasy plant mini kit (Qiagen GmbH, Hilden, Germany) as described in the manufacturer’s instructions. DNA quality and quantity were spectro-photometrically confirmed using NanoDrop (Thermo Fischer Scientific, Darmstadt, Germany). The qPCR assays were run using the CFX96 Touch™ Real-Time PCR Detection System (BioRad, München, Germany). For absolute quantification of the three microorganisms, a standard curve was generated using serial dilutions of genomic DNA isolated from *S. indica*, *K. radicincitans*, and *C. fungivorans*, respectively. To establish the relationship between C_q_-values and the concentration of target fungal or bacterial DNA (ng/µL), a calibration curve was established for each primer combination using a dilution series of the corresponding DNA as a template.

Standard curves were generated by plotting threshold cycles (C_q_) versus genome equivalents of the microorganism, as described by Whelan et al. [89]. C_q_ values in each dilution (dilution factor 10) were measured in triplicate using real-time qPCR to generate the standard curves for *S. indica*, *K. radicincitans*, and *C. fungivorans,* respectively. The C_q_ values were plotted against the logarithm of their initial template copy numbers. Each standard curve was generated by a linear regression of the plotted points (Appendix A). From the slope of each standard curve, PCR amplification efficiency (E) was calculated as described by Rasmussen et al. [90].

*S. indica* was quantified using a SYBR green I-based method using primers targeted to *Pitef1* [91] (Appendix A). The SsoAdvanced universal SYBR green super mix (Biorad, Munich, Germany) was used at a final concentration of 1× for the PCR. Primers Tef-f and Tef-r were added to a final concentration of 400 nM along with 1 µL of DNA template. The volume of the reaction mixture was adjusted to 20 µL using nuclease-free water. Amplifications were performed using the following conditions: 5 min at 95 °C, followed by 40 cycles of 30 s at 95 °C and 30 s at 59 °C.

*C. fungivorans* was quantified using a TaqMan-based method using the primer pair EddyFor and Eddyrev along with *Collimonas*-specific probe Sophie (Appendix A) as described [45]. SensiFAST Probe lo-ROX Mix (Bioline, Luckenwalde, Germany) was used at a final concentration of 1× for PCR. Primers Eddy3for and Eddy3rev were added to a final concentration of 400 nM along with TaqMan probe Sophie (100 nM) and 1 µL of DNA template. The volume of the reaction mixture was adjusted to 20 µL using nuclease-free water. Amplifications were performed using the following conditions: 5 min at 95 °C, followed by 45 cycles of 10 s at 95 °C and 45 s at 60 °C.

Lastly, *K. radicincitans* was quantified using a TaqMan assay with the primer pair 519f and E.radr along with the E.rad TaqMan probe specific to *Kosakonia* (Appendix A) as described by Ruppel et al. [92]. Briefly, SensiFAST Probe lo-ROX Mix (Bioline) was used at a final concentration of 1× for PCR. Primers 519f and E.radr were added to a final concentration of 400 nM along with the E.rad TaqMan probe (100 nM) and 1 µL of DNA template. The volume of the reaction mixture was adjusted to 20 µL using nuclease-free water. Amplifications were performed using the following conditions: 5 min at 95 °C, followed by 45 cycles of 10 s at 95 °C and 30 s at 60 °C.

### 2.11. Confocal Laser Scanning Microscopy (CLSM) Analysis

The colonization of *S. indica* hyphae by *K. radicincitans* and *C. fungivorans* was observed using a laser confocal microscope 510 META (Carl Zeiss Jena GmbH, Jena, Germany) and a Zeiss Plan-Apochromat 63×/1.4 oil objective or 25× or 40× objectives. GFP-expressing strains of *K. radicincitans* and *S. indica* were used to visualize the interaction, whereas the mCherry strain of *C. fungivorans* was used to visualize its interaction with *S. indica*. The RFP and GFP-expressing strains were produced by the plasmid insertion method [74,92,93]. *K. radicincitans* and *S. indica* eGFP fluorescence signals were captured using argon laser excitation at 488 nm (BP 505–50), whereas *C. fungivorans* RFP fluorescence signals were captured using argon laser excitation at 543 nm (BP 530-600).

To check the viability of *S. indica*, when the fungus was co-cultured with either *C. fungivorans* or *K. radicincitans*, the Live/dead^®^ Baclight^TM^ bacterial viability kit (Thermo Fisher Scientific) was used. Briefly, mycelia were incubated with a mixture containing equal volumes of Syto9 (3.34 mM) and propidium iodide (PI) (20 mM) for 20 min prior to the analysis. PI emits red fluorescence when bound to nucleic acids and impaired cell membranes. Therefore, PI is excluded from viable cells and can only penetrate cells when their membrane is compromised, thus it can be used to identify dead cells [93]. The images were analyzed using ZEN (blue edition) software (Carl Zeiss Jena GmbH, Germany).

### 2.12. Data Analysis

Statistical analysis was performed using STATISTICA version 13 software (TIBCO Statistica^®^ 13.3.0). Experiments in Section 2.1, Section 2.2, Section 2.3, Section 2.4, Section 2.5, Section 2.6, Section 2.7 and Section 2.8 were analyzed by ANOVA, followed by a Tukey’s range test or Tukey’s test of additivity with a cut-off significance at *p* ≤ 0.05.

## 3. Results

### 3.1. Fungal Growth Inhibition Assays

The in vitro assay performed to check fungal growth inhibition by the two bacteria revealed that both *C. fungivorans* and *K. radicincitans* did not inhibit the growth of *S. indica* under the tested experimental conditions (Appendix A).

### 3.2. Mycophagy Assay

The results of the mycophagy assays indicate that both *K. radicincitans* and *C. fungivorans* could utilize the fungus as a source of carbon (Figure 1a). However, there was a significant difference in the growth of the two bacteria when confronted with *S. indica*. The growth of *K. radicincitans* increased by only 2-fold when confronted with *S. indica* as the only source of nutrients. Whereas, the growth of *C. fungivorans* increased by 8-fold (Figure 1a) indicating it could benefit more from *S. indica* than *K. radicincitans*. Interestingly, the biomass of *S. indica* was reduced by 2.5-fold when confronted with *C. fungivorans* (Figure 1b). Whereas in the case of *K. radicincitans* the fungal biomass was reduced by only 1.6-fold (Figure 1b). Therefore, though both bacteria could benefit from the fungus, they did so in different quantities. When the hyphae of *S. indica* were visualized microscopically (Figure 2a), no structural damage to the fungal hyphae was detected when it was confronted with *K. radicincitans* (Figure 2b). However, in the case of *C. fungivorans* the increased bacterial growth was coupled with broken fungal hyphae (Figure 2c). Therefore, we hypothesized that the two bacteria use different strategies to benefit from *S. indica*.

### 3.3. Growth of the Bacteria on Fungal Metabolites

In order to test the hypothesis of different feeding strategies by *C. fungivorans* and *K. radicincitans*, we evaluated the growth of both bacteria in minimal (M9) medium supplemented with *S. indica* metabolites as the only source of carbon. The control treatment consisted of metabolites extracted from the media without fungus. During 48 to 72 h of cultivation, both bacteria showed a significant increase in biomass in the media amended with fungal metabolites compared to the control treatment (Figure 3a,b). However, the growth of *K. radicincitans* increased by 2-fold, whereas the growth of *C. fungivorans* increased by only 1.3-fold. This suggests that while both bacteria could utilize the fungal metabolites for their growth, *K. radicincitans* was more efficient in feeding on fungal metabolites compared to *C. fungivorans* (Figure 3a).

### 3.4. Co-Cultivation of Bacteria and Fungus (with or without Physical Contact)

In order to further confirm the different feeding strategies used by the two bacteria, we evaluated the role of physical attachment between the two microorganisms. We hypothesized that since *C. fungivorans* can benefit from the fungus via hyphal feeding, physical attachment to the fungus would be crucial for it to benefit from *S. indica*. In the case of *K. radicincitans*, physical attachment could facilitate bacterial access to fungal metabolites but the bacteria could also benefit from the fungus without any physical attachment.

*S. indica* was co-cultivated with either *K. radicincitans* or *C. fungivorans* in 6-well plates. When *S. indica* was physically separated from the bacteria, no indication for the growth of *C. fungivorans* was observed when it was co-cultivated with the fungus or alone (Figure 4a), confirming that in order to benefit from *S. indica*, physical attachment to the fungus is a prerequisite for *C. fungivorans*. In the case of *K. radicincitans*, the bacterial growth increased by 6.6-fold even when it was physically separated from *S. indica* compared to the bacteria-only treatment (Figure 4a). Therefore, physical attachment to *S. indica* is not crucial for *K. radicincitans*. When present in physical contact with the fungus, the growth of *C. fungivorans* increased by 22.5-fold compared to the bacteria without *S. indica* (Figure 4b). Interestingly, the growth of *K. radicincitans* increased by 29-fold compared to the treatment without fungus (Figure 4b).

### 3.5. Tripartite Interaction: K. radicincitans, S. indica, and C. fungivorans

In order to test the hypothesis that *K. radicincitans* can protect *S. indica* against the fungus-feeding bacterium *C. fungivorans*, we performed an in vitro co-cultivation assay. First, when *S. indica* was cultivated with *K. radicincitans*, the bacteria formed a close physical association with the fungus, leading to the formation of a thick biofilm, which colonized the fungal hyphae (Figure 5a). Afterwards, when this pre-assembled *S. indica–K. radicincitans* biofilm was confronted with *C. fungivorans,* the fungus covered a significantly larger area on the Petri dish compared to the control, which consisted of *S. indica* without *K. radicincitans* (Figure 5c,d). In the control treatment, *S. indica* did not show any growth beyond the zone of confrontation with *C. fungivorans* (Figure 5d, red arrow). Additional images are available in the Appendix A.

### 3.6. Quantification of Bacterial and Fungal Copy Numbers via qPCR

The tripartite interaction between *S. indica*, *K. radicincitans,* and *C. fungivorans* was studied in vitro using a 6-well plate system comprising of buffered liquid medium. The fungal single gene copy number was estimated using qPCR. Two hours post inoculation, the fungal single gene copy number did not show any change when the fungus was co-cultivated with *K. radicincitans* or *C. fungivorans* or both compared to the uninoculated fungus control. However, a significant decrease in the *S. indica* single gene copy number was observed 72 h post inoculation when it was co-cultured with *C. fungivorans* and when *S. indica* was co-cultured with a combination of *C. fungivorans* and *K. radicincitans,* compared to the uninoculated fungus control (Figure 6a). In the same experiment, the single gene copy numbers of the two bacteria were also quantified in the presence of *S. indica*. Interestingly, the presence of *S. indica* significantly increased the single gene copy number of *K. radicincitans* 2 h post inoculation compared to C. *fungivorans*. A similar trend was seen after 72 h, indicating that *S. indica* supported the growth of *K. radicincitans* more than the growth of *C. fungivorans* (Figure 6b). We also quantified each bacterium when all three microorganisms were present in the system. Interestingly, after 2 h the single gene copy number of *K. radicincitans* was significantly higher than *C. fungivorans* when both bacteria were present in the system. However, this trend disappeared after 72 h probably due to intense competition for nutrients.

### 3.7. Confocal Laser Scanning Microscopy (CLSM) Analysis

Live/dead staining:

Live/dead staining revealed structural damage to the hyphae of *S. indica* when it was confronted with *C. fungivorans*, which was evident by enhanced red fluorescence (Figure 7c). In the case of *K. radicincitans*, few hyphae of *S. indica* were found with a damaged cell wall (Figure 7d) compared to the control treatment (Figure 7a). Heat-killed hyphae were used as positive controls and showed increased red fluorescence due to PI staining (Figure 7b).

Interaction of *S. indica* with *C. fungivorans* and *K. radicincitans*

When *S. indica* was cultured without any bacteria, no damage to fungal hyphae could be observed (Figure 8a). When *S. indica* was co-cultured with the RFP-expressing strain of *C. fungivorans,* bacterial aggregates were found attached to the hyphal tips of the fungus 4 h post inoculation (Figure 8b). Lysis of hyphal tips was evident after 24 h (Figure 8c) and 48–72 h post inoculation, broken hyphae of *S. indica* were observed (Figure 8d,e). When *S. indica* was co-cultivated with the GFP-expressing strain of *K. radicincitans*, the bacteria colonized the fungal hyphae leading to formation of bacterial aggregates during the 72 h period of incubation and no lysis of fungal hyphae was detected microscopically (Appendix A).

## 4. Discussion

In the present study, we provide evidence indicating that the hyphae-colonizing bacterium *Kosakonia radicincitans* can confer protection to the beneficial root-endophytic fungus *Serendipita indica* against the fungus-feeding bacterium *Collimonas fungivorans* when co-cultured on solid agar medium. The basis of this protective effect seems to be based on the principle of reciprocity, whereby the bacteria utilize the fungal metabolites for growth and in exchange provide protection to the fungus against *C. fungivorans* by forming a thick biofilm. *K. radicincitans* harbors an arsenal of genes encoding for biofilm formation and has been reported to establish biofilms on plant roots for successful colonization [63,94]. We present here evidence of biofilm formation by this bacterium on the hyphae of the plant-beneficial fungus *S. indica* for the first time. However, when the three microbes were co-cultivated in liquid medium using the 6-well plate system, this protective effect diminished. The probable reason for this could be the extreme nutrient-limiting conditions of the buffered system. Furthermore, it is possible that *K. radicincitans* biofilms were not properly established due to the short duration of this experiment. There has been previous work on the protective effect of bacterial biofilm formation against antifungal compounds such as cycloheximide [95]. In this study we go several steps further, and provide evidence that *S. indica* can grow while it is being protected from mycophagous bacterial feeding by a *K. radicincitans* biofilm along its hyphae. This is a particularly interesting finding and requires further investigation as it indicates a mutually beneficial interaction between two economically important plant growth-promoting microorganisms.

Interestingly, in our experiments the interaction of *S. indica* with *K. radicincitans* and *C. fungivorans* revealed that the two bacteria use different feeding strategies to benefit from the fungus, namely, metabolite feeding and hyphal feeding (Figure 9). *C. fungivorans* could only grow when it was physically associated with the hyphae of *S. indica,* thereby indicating that physical attachment to the fungus is probably a prerequisite for *C. fungivorans* to benefit from it. This close proximity to *S. indica* might be important for *C. fungivorans* as it can trigger the production of specific metabolites [33,96,97], as was reported in the case of *Aspergillus nidulans* and its hyphae-adhering bacterium *Streptomyces hygroscopicus* [98]. In addition to this, close association to the fungal hyphae can also lead to easy and increased access to fungus-derived nutrients, probably by breaking the fungal hyphae to access the cytoplasmic contents [47,96,97]. However, this close physical association had a detrimental effect on the fungus, as was revealed during our microscopy analysis, which showed that *C. fungivorans* could colonize the hyphal tips of *S. indica*, leading to the formation of bacterial aggregates which eventually led to the breakage of fungal hyphae. This is in line with the previously reported mycophagous lifestyle of *C. fungivorans* [41,84]. In a study by Höppener and colleagues [50], inoculation of a soil microcosm containing field soil with fungus belonging to *Absidia* sp. led to a significant increase in the population of indigenous collimonads, however no reduction in the fungal biomass was reported. In addition to this, in the same experiment, no increase in the population of common fungal hyphae-associated bacteria such as *Pseudomonas* and *Burkholderia,* which are known to feed on fungal metabolites, was detected [38,53,99,100]. This observation further supports the “hyphal feeding” strategy of collimonads. In contrast to this study, our results indicate a significant reduction in the biomass of *S. indica* when it was co-cultivated with *C. fungivorans* under in vitro conditions. This could be due to the differences in sensitivity of the two fungi towards collimonads or the feeding preference of the bacteria [41,45].

Contrary to *C. fungivorans*, physical attachment to the fungus was not necessary for *K. radicincitans* to benefit from *S. indica* but it improved the growth of bacteria probably due to the higher diffusion of nutrients. Our microscopy analysis revealed that *K. radicincitans* colonized the hyphae of *S. indica,* forming stable biofilms, but no damage to the fungal hyphae could be detected in the tested experimental conditions. This indicates that *K. radicincitans* could benefit from *S. indica* by feeding on the fungal metabolites and, unlike *C. fungivorans,* did not cause lysis of the fungal hyphae. The role of *K. radicincitans* as a plant growth-promoting rhizobacterium has been extensively studied [59,60,101,102], but the interaction of this bacterium with common soil fungi has been poorly elucidated [63]. Biofilm formation by *K. radicincitans* on the hyphae of *S. indica* can have several advantages for the bacteria. First, the attachment to fungal hyphae as a substratum can lead to better access to fungal exudates and limit the competition for space and nutrients with other soil bacteria [103]. Second, *K. radicincitans* could use the water film along the fungal hyphae to migrate in the soil to nutrient-rich niches [30,104,105,106]. In addition to this, bacterial biofilms can confer protection to the inhabiting bacteria against grazing by soil protozoans compared to their planktonic state [107,108]. An interesting observation among fungus-colonizing bacteria is the essential role of motility in the colonization and movement along the fungal hyphae via swimming/gliding in the associated water film [104,105,106]. In the case of *K. radicincitans*, it has been reported to be a highly mobile and competitive rhizospheric bacterium and this is probably due to the genomic adaptation of the bacterium, as *K. radicincitans* harbors genes of T3SS clustered together with chemotaxis genes and all other genes required for flagella biosynthesis, suggesting that this strain uses “T3SS” genes for motility purposes [63]. Therefore, it is important to investigate the growth of *K. radicincitans* along the fungal hyphae of different taxonomical groups of fungi and artificial glass fibers or nylon threads as a proxy for fungal hyphae to evaluate if the colonization of fungal hyphae is a rather general phenomenon for *K. radicincitans* or was it regulated by the phenomenon of reciprocity in the case of *S. indica*.

“Cooperation” between fungi and bacteria with reciprocal benefits has been described extensively for mycorrhizal fungi [20,38,104,109,110]. In addition to protective services, such “mycorrhiza helper bacteria” can also support the fungal partner by providing nutrients that are otherwise inaccessible to the fungus. Zhang and colleagues [20] showed that a phosphate-solubilizing bacterium, *Rahnella aquatilis* (strain HX2), colonized the hyphae of the arbuscular mycorrhizal (AM) fungus, *Rhizoglomus irregulare,* and provided phosphorous to the fungus from an organic P source that is otherwise inaccessible to the fungus in exchange for fungal-derived carbon. Later on, Jiang et al. [104] concluded that *R. aquatilis* (strain HX2) formed a biofilm on the AM fungal hyphae which aided in the swimming of the bacteria in a water film along the fungal hyphae while feeding on the fungal exudates and directing the fungus towards the phytate, thereby confirming the theory of cooperation via reciprocity. On similar lines, the interaction between *S. indica* and the nitrogen-fixing bacterium, *Azotobactor chroococcum,* revealed a close physical association between the two microbes, which enhanced the carbon pool in the mycelium and triggered nitrogen influx [72]. *S. indica* lacks nitrate transporters, nitrate, and nitrite reductases, but can readily take up ammonia [111]. It is therefore possible that, in addition to protection, *K. radicincitans,* which is able to fix atmospheric nitrogen even under additional nitrogen sources [58], could probably also provide nitrogen to *S. indica* in exchange for fungal-derived carbon. Therefore, it is important to further test the reciprocal benefits of the *K. radicincitans–S. indica* interaction. One approach could be to trace nutrient exchange between the microorganisms by applying a dual isotope labeling technique, using ^15^N-labeled KNO_3_ and ^13^C labelled glucose in the in vitro co-culturing system developed in this study [112]. In addition to this, the expression of two high-affinity ammonium transporters of *S. indica,* namely PiAMT1 and PiAMT2, which are predicted to be involved in the transport of ammonium, can also be monitored during co-cultivation with *K. radicincitans* to elucidate the molecular mechanism of this interaction [113].

*S. indica* and *K. radicincitans* are important bioinoculants for which commercial products are already available. From an application point of view, the biggest challenge for microbial inoculants to enhance crop productivity and yield in low-input agricultural systems are rhizosphere establishment, plant colonization, successful competition with native microorganisms, and maintenance of viability in the field [113]. It is often observed that beneficial activity shown by the microbes in laboratory assays does not translate to effective outcomes in the field. This has been attributed to effects of abiotic as well as biotic factors [114,115]. Additionally, antagonism by other microorganisms can have a negative effect on the viability and efficacy of microbial inoculants [115,116]. Bacterial-fungal biofilmed biofertilizers (BFBBs) are being extensively tested to enhance the efficacy of microbial inoculants under field conditions [117,118]. The biofilms can provide structure and stability to the partners, thereby enhancing the functionality of the beneficial microbes [119,120]. The use of biofilmed inocula has been shown to protect the bio-inoculant against predators such as earthworms [121]. However, the ubiquitous presence of mycophagous bacteria, belonging to diverse bacterial taxa, might lead to survival pressure on the beneficial microbes. Therefore, it would be important to test the survival of fungal inoculants with biofilms of different bacteria against potential mycophagous bacteria to improve the efficacy of bioinoculants in the field. The first step towards achieving this goal could be to conduct greenhouse experiments with controlled environmental conditions to monitor this protective effect under soil conditions. Therefore, more experiments are needed to confirm the outcome of this bacteria–fungal–bacteria tripartite interaction in planta—especially from the point of view of application.

## 5. Conclusions

In summary, our findings demonstrate that biofilm formation by *K. radicincitans* protected the hyphae of plant-beneficial fungus *S. indica* against the mycophagous bacterium *C. fungivorans* on solid agar medium. However, the tripartite interaction between *S. indica*, *C. fungivorans*, and *K. radicincitans* has a distinct cost-benefit relation for all three partners. The two bacteria differed in their mechanism to benefit from the fungus. In the case of *K. radicincitans,* the bacteria could actively feed on fungal metabolites, leading to increased bacterial biomass, with or without any physical contact with the fungus. However, in the case of *C. fungivorans,* physical attachment to the fungal hyphae was crucial for the bacteria to benefit from the fungal partner. The increase in bacterial biomass was directly related to the reduced fungal biomass, providing evidence for the mycophagous lifestyle of such bacteria as previously reported. Our study also demonstrates an innovative approach to the study of complex tripartite interaction using different in vitro systems, which might pave the way for further screening of beneficial bacterial-fungal interactions. In future studies, it will be crucial to estimate the loss of beneficial fungi due to the feeding of mycophagous bacteria. Since biofilm formation on fungal hyphae seems to be a more common phenomenon than previously reported, it is imperative to evaluate the protective effect of such bacterial-fungal biofilms against bacterial predators [21,100]. Screening of “potential protective” bacteria can aid in developing more efficient BFBBs. Such biofilmed inocula might enhance the survival and efficacy of bio-inoculants in field conditions. More studies are required to evaluate the endurance of such biofilmed consortia of bacteria and fungi against bacterial predators in soil systems during greenhouse and field trials.

## Figures and Tables

**Figure 1 microorganisms-09-01566-f001:**
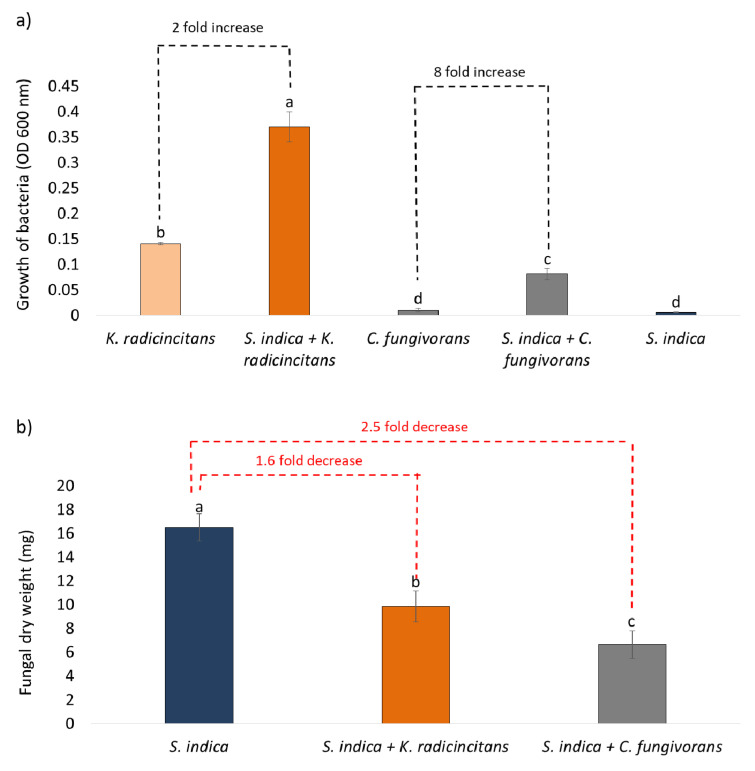
Results of mycophagy assays. (**a**) Growth of *Collimonas fungivorans* and *Kosakonia radicincitans* when co-cultivated with *Serendipita indica* as the sole source of carbon. (**b**) Fungal biomass of *S. indica* when co-cultivated with either *C. fungivorans* or *K. radicincitans*, or the uninoculated fungus (control). Different letters above bars represent significant differences (*p* < 0.05) according to one-way ANOVA followed by a Tukey’s range test. Error bars indicate the standard deviation of means of four different replicates.

**Figure 2 microorganisms-09-01566-f002:**
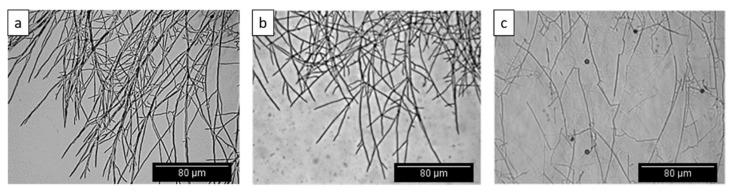
(**a**) Morphology of the uninoculated *S. indica* hyphae (control). (**b**) Fungal morphology in the presence of *K. radicincitans.* (**c**) Fungal morphology in the presence of *C. fungivorans*.

**Figure 3 microorganisms-09-01566-f003:**
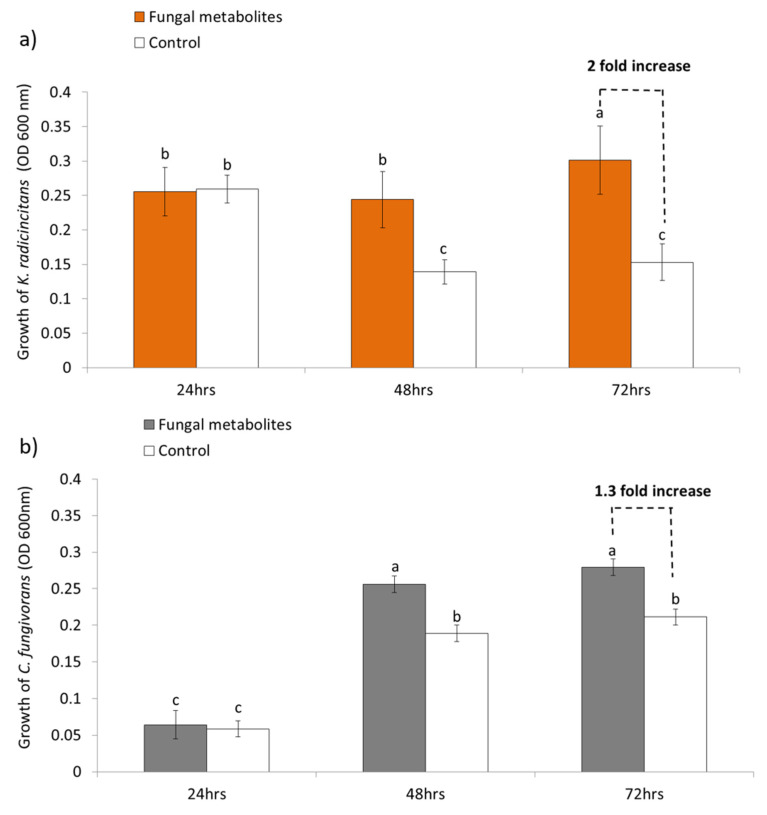
Growth of (**a**) *K. radicincitans* and (**b**) *C. fungivorans* when cultivated in M9 medium amended with metabolites extracted from complete medium (CM) inoculated with *S. indica* after two weeks of incubation. White bars represent the growth of *K. radicincitans* and *C. fungivorans* in the control treatments amended with metabolites extracted from CM medium without *S. indica*. OD_600_ nm was measured in all the treatments after 24 h, 48 h, and 72 h. Different letters above bars represent significant differences (*p* < 0.05) according to two-way ANOVA using treatment (*S. indica* metabolites or CM without fungus) and time (2 h and 72 h) as factors; followed by a Tukey’s additive test. Error bars indicate the standard deviation of means of twelve different replicates.

**Figure 4 microorganisms-09-01566-f004:**
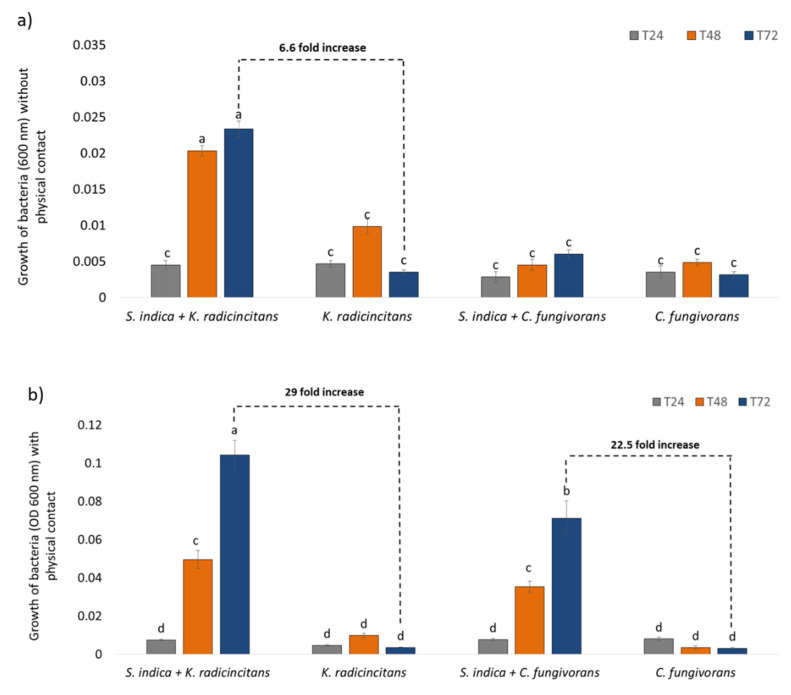
(**a**) Growth of *C. fungivorans* and *K. radicincitans* when co-cultivated without physical contact with *S. indica.* (**b**) Growth of *C. fungivorans* and *K. radicincitans* when co-cultivated with physical contact with *S. indica*. Different letters above bars represent significant differences (*p* < 0.05) according to three-way ANOVA with bacteria (*C. fungivorans* or *K. radicincitans*), fungus (with *S. indica* or without *S. indica*), and time (2 h and 72 h) as factors followed by a Tukey’s additive test. Error bars indicate the standard deviation of means of six different replicates.

**Figure 5 microorganisms-09-01566-f005:**
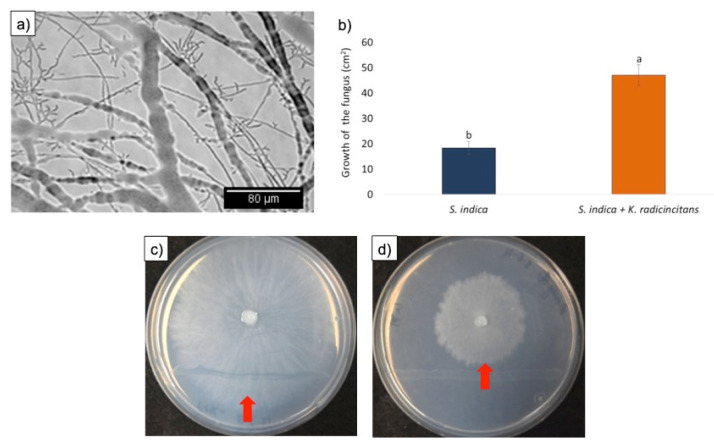
(**a**) *K. radicincitans* forming a biofilm-like structure on the hyphae of *S. indica.* (**b**) The area covered by uninoculated *S. indica* and *S. indica*–*K. radicincitans* consortia when confronted with the fungus-feeding bacterium *C. fungivorans. (***c**) A Petri dish showing the growth of *S. indica–K. radicincitans* consortia (red arrow) when confronted with *C. fungivorans.* (**d**) Growth of uninoculated *S. indica* (red arrow) when confronted with *C. fungivorans*. Different letters above bars represent significant differences (*p* < 0.05) according to one-way ANOVA followed by a Tukey’s range test. Error bars indicate the standard deviation of means of six different replicates.

**Figure 6 microorganisms-09-01566-f006:**
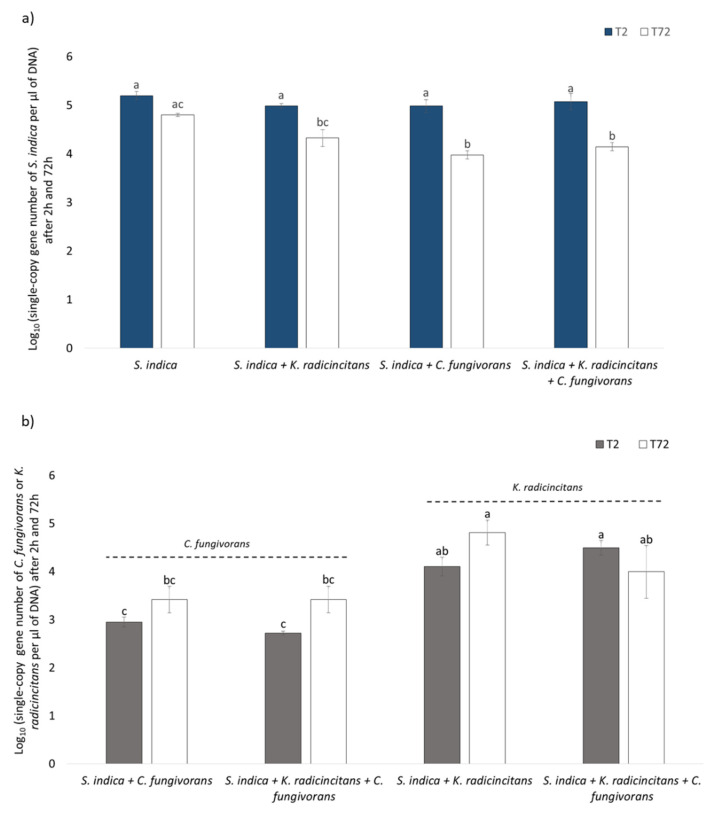
(**a**) Average log_10_ transformed single-copy gene numbers of *S. indica* (per µL of DNA) when co-cultivated with *K. radicincitans*, *C. fungivorans,* or both, 2 h and 72 h post inoculation. (**b**) Average log_10_ transformed single-copy gene numbers of *C. fungivorans* and *K. radicincitans* (per µL of DNA) when co-cultivated with S*. indica*, 2 h and 72 h post inoculation. Different letters above bars represent significant differences (*p* < 0.05) according to two-way ANOVA, using treatment and time as factors followed by a Tukey’s range test. Error bars indicate standard deviation of means of six different replicates.

**Figure 7 microorganisms-09-01566-f007:**
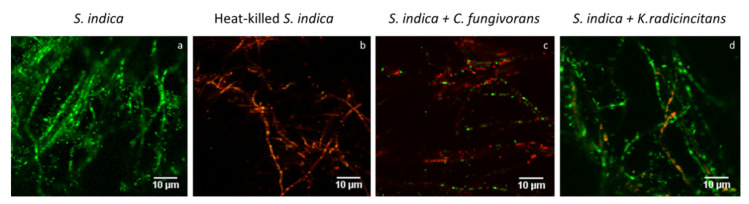
Imaging of microcolonies of *S. indica* co-cultured with *K. radicincitans* or *C. fungivorans* for 72 h at 24 °C. The samples were dual-stained with propidium iodide (red) and Syto9 (green) and observed under confocal microscope at 65× magnification. (**a**) *S. indica* hyphae without bacteria (control). (**b**) Heat-killed hyphae of *S. indica* without bacteria (control). (**c**) *S. indica* hyphae when co-cultured with *C. fungivorans.* (**d**) *S. indica* hyphae when co-cultured with *K. radicincitans*.

**Figure 8 microorganisms-09-01566-f008:**
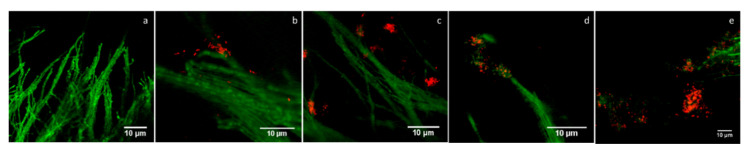
Imaging of microcolonies of *S. indica* co-cultured with *C. fungivorans* for 72 h at 24 °C. The colonization of the fungal hyphae by *C. fungivorans* was observed microscopically. (**a**) *S. indica* hyphae without bacteria (control). (**b**) *C. fungivorans* attached to the hyphal tips and forming bacterial aggregates 4 h post inoculation. (**c**) Lysis of hyphal tips was observed after 24 h. (**d**,**e**) After 72 h, broken hyphae of *S. indica* were observed. The GFP-and RFP-expressing strains of *S. indica* and *C. fungivorans* were used and images were taken using a confocal microscope at 40× magnification.

**Figure 9 microorganisms-09-01566-f009:**
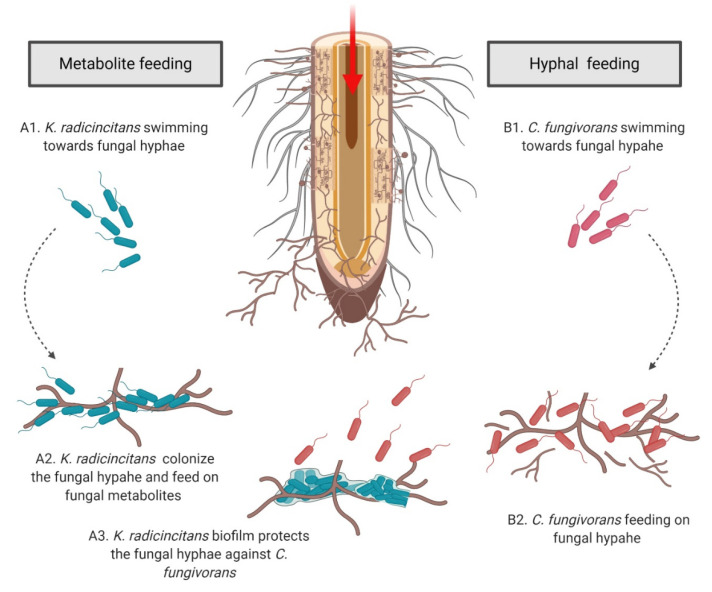
A conceptual model of hyphal protection and bacterial feeding strategies employed by *K. radicincitans* and *C. fungivorans*. Photosynthetically fixed carbon enters the rhizosphere via roots and fungal hyphae. Metabolite feeding: *K. radicincitans* colonizes the fungal hyphae and forms a thick biofilm while feeding on fungal metabolites. The biofilm confers protection to the fungal hyphae against *C. fungivorans* on solid agar medium (A1–A3). Hyphal feeding: the fungus-feeding bacterium, *C. fungivorans,* colonizes and feeds on the fungal hyphae via mycophagy (B1–B2).

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
