# Peer review of "It Takes Two to Tango: A Bacterial Biofilm Provides Protection against a Fungus-Feeding Bacterial Predator"

_microorganisms, 2021, doi:10.3390/microorganisms9081566_

Round 1
Reviewer 1 Report
See attached .pdf.

Author Response
We would like to thank the reviewer for constructive comments and suggestions. Those have substantially improved the manuscript.
For the replies, see the attached file.

Reviewer 2 Report
The authors further improved the work. For this, I recommend the publication.
Author Response
We thank the reviewer for the kind words.
Reviewer 3 Report
The revised Manuscript is ready to publish.
Author Response
We thank the reviewer for the positive evaluation of our work.
This manuscript is a resubmission of an earlier submission. The following is a list of the peer review reports and author responses from that submission.
Round 1
Reviewer 1 Report
See the attached .pdf.

Reviewer 2 Report
The article entitled: "Bacterial biofilm provides protection against fungus feeding bacterial predator" is well presented and structured.
In this study, the authors provide evidence indicating that the hyphae colonizing bacterium Kosakonia radicincitans can confer protection to the beneficial root endophytic fungus Serendipita indica against the fungus feeding bacterium Collimonas fungivorans when co-cultured on solid agar medium. The basis of this protective effect seems to be based on the principle of reciprocity. This is an interesting topic to develop from an application point of view to obtain effective outcomes on crop protection.
The scientific method described is correct, and all the experiments have been carried out propriety. Also, I congratulate the authors for the chosen title; it's really spot on.
Considering these premises, I recommend the paper for publication in the present form.
Reviewer 3 Report
The current manuscript discribed the fungus-bacterium interactions by using in vitro systems combined with Confocal Laser Scanning Microscopy and Real Time Quantitative PCR. The results concluded that a nitrogen-fixing bacterium Kosakonia radicincitans can provide protection to the plant beneficial fungus Serendipita indica, which inhabits the rhizosphere and colonizes the plants as an endophyte, from the fungus feeding bacterium Collimonas fungivorans. This manuscript contains interesting results that can help the readers to understand the bacterial and fungal interaction.